# ADAPTIVE SPEECH DURATION MODIFICATION USING A DEEP-GENERATIVE FRAMEWORK

## ABSTRACT

We propose the first method to adaptively modify the duration of a given speech signal. Our approach uses a Bayesian framework to define a latent attention map that links frames of the input and target utterances. We train a masked convolutional encoder-decoder network to generate this attention map via a stochastic version of the mean absolute error loss function. Our model also predicts the length of the target speech signal using the encoder embeddings, which determines the number of time steps for the decoding operation. During testing, we generate the attention map as a proxy for the similarity matrix between the given input speech and an unknown target speech signal. Using this similarity matrix, we compute a warping path of alignment between the two signals. Our experiments demonstrate that this adaptive framework produces similar results to dynamic time warping, which relies on a known target signal, on both voice conversion and emotion conversion tasks. We also show that the modified speech utterances achieve high user quality ratings, thus highlighting the practical utility of our method.

## 1 INTRODUCTION

Human speech is a rich and varied mode of communication that encompasses both language/semantic information and the mood/intent of the speaker. The latter is primarily conveyed by prosodic features, such as pitch, energy, speaking rate, and voice quality. There are many applications where understanding and manipulating these prosodic features is required. Consider voice conversion systems as an example. Pitch and energy modifications are used to passively inject emotional cues into the neutral speech or to change the overall speaking style (A. Russell et al., 2003; Schacter et al., 2011; Shankar et al., 2019a;b; Valle et al., 2019). Prosodic features are also used to evaluate the quality/engagement in human machine dialog systems (Swerts & Krahmer, 2000), and they play a significant role in speaker identification and recognition systems (Park et al., 2016).

While there are many approaches for automated pitch and energy modification (Toda et al., 2007; Aihara et al., 2012; Kaneko & Kameoka, 2017; Shankar et al., 2020b;a), comparatively little progress has been made in changing the duration/speaking rate of an utterance. In fact, the speaking rate plays a crucial role in conveying emotions (Schmidt et al., 2016) and in diagnosing human speech pathologies (Bayerl et al., 2020). The speaking rate is difficult to manipulate because, unlike pitch or energy, there is no explicit coding for either the signal duration or speaking rhythm. Rather, these features are implicitly defined by the cardinality of the set of frames over a particular interval of interest. This cardinality is a global parameter that masks subtle variations in the speaker rate over an utterance. As a result, duration modification algorithms are not adaptive. Instead, they either require considerable user supervision or they are geared towards aligning to known speech signals.

Perhaps the earliest duration modification method is the time-domain pitch synchronous overlap and add (TD-PSOLA) algorithm (Charpentier & Stella, 1986). TD-PSOLA modifies the pitch and duration of a speech signal by replicating and interpolating between individual frames centered at the peaks of auto-correlation signal. However, the user must manually specify both the portion of speech to modify and the exact manner in which it should be altered. Hence, the method is neither automated nor adaptive. An alternative approach is dynamic time warping (DTW), which finds the optimal time alignment between two parallel speech utterances (dtw, 2008). DTW constructs a pairwise similarity matrix between all frames of the two utterances and estimates a *warping path* between the starting $(0, 0)$ and ending $(T_s, T_t)$ points of the utterances based on a Viterbi-like de-

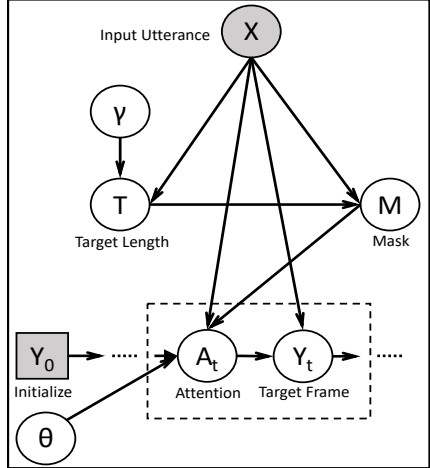

Figure 1: Graphical model for duration modification. $\gamma$ and $\theta$ are the model parameters inferred during training. Attention $A_t$ is conditionally independent of target length $T$ given $X$ and $M$

coding of the similarity matrix. While simple, DTW requires both the source and target utterances to be known *a priori*. Hence, it cannot be used for on-the-fly modification of new signals.

Finally, recent advancements in deep learning have led to a new generation of neural vocoders that disentangle the semantic content from the speaking style (Oord et al., 2016; Shen et al., 2017; Wang et al., 2017). These vocoders can alter the speaking rate via the learned style embeddings. While these models represent seminal contributions to speech synthesis, the latent representations are learned in an unsupervised manner, which makes it difficult to control the output speaking style in predictable manner. Another drawback of these methods is the large amount of data and computational resources required to train the models and generate new speech (Yasuda et al., 2020).

In this paper, we introduce the first fully-automated adaptive speech duration modification scheme. Our approach combines the representation capabilities of deep neural networks with the structured simplicity of dynamic decoding. Namely, we model the alignment between a source and target utterance via a latent attention map; these maps are used as replacement of similarity matrix for backtracking. We train a masked convolutional encoder-decoder network to estimate these attention maps using a stochastic mean absolute error (MAE) formulation. Unlike the conventional DTW (dtw, 2008) algorithm, once trained our framework operates entirely on the source utterance without needing to reference the target. We demonstrate our framework on a voice conversion task using the CMU-Arctic dataset (Kominek & W Black, 2004) and on three multi-speaker emotion conversion tasks using the VESUS dataset (Sager et al., 2019). Our experiments confirm that the proposed model can perform open-loop duration modification and produces high-quality speech.

## 2 METHOD

Fig. 1 illustrates our underlying generative process. Given an utterance $X$, we first estimate the length $T$ of the (unknown) target utterance $Y$ and subsequently use it to estimate a mask $M$ for the attention map. The mask restricts the domain of the attention vectors $A_t$ at each frame $t$ during the inference stage to mitigate distortion of the output speech. We use paired data $(X_{tr}, Y_{tr})$ to train a convolutional encoder-decoder network to generate the attention vectors. During testing, we first generate the attention map from the input $X$ and use it to produce the target speech $Y$.

### 2.1 LOSS FUNCTION

Formally, let $X \in \mathbb{R}^{D \times T_s}$ denote the input speech. In this work, $X$ corresponds to the Mel filter-bank energies extracted from short-time moving window analysis, where $D$ is the number of filter-banks, and $T_s$ is the number of temporal frames in the utterance. Similarly, we denote the target speech as $Y \in \mathbb{R}^{D \times T}$. Notice that the target utterance length $T$ is usually different from $T_s$.

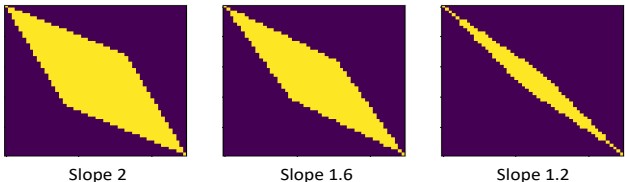

Figure 2: Binary attention masks with 3 different slopes.

Our generative process for a single frame of the target speech is represented as follows:

$$T \sim \text{Laplace}(T^0, b_T) \quad and \quad Y_t \sim \text{Laplace}(Y_t^0, b_y), \tag{1}$$

where $T$ is the estimated length of the target utterance, and $Y_t$ is the target Mel filter-bank energy features at time $t$. The parameters $\{T^0, b_T, Y_t^0, b_y\}$ of the distributions are unknown and implicitly estimated via a deep neural network. The neural network is parameterized by $\gamma$ and $\theta$ (Fig. 1).

By treating the unknown parameters as functions of the input $X$, we obtain the following estimating equations for the target sequence length and frame-wise Mel filter-bank energies:

$$\hat{T} = f_\gamma(X) \quad and \quad \hat{Y}_t = X \cdot A_t + f_\theta(X, \hat{Y}_{0:t-1}). \tag{2}$$

The functions $f_\gamma(\cdot)$ and $f_\theta(\cdot, \cdot)$ correspond to the length prediction and energy estimation component of the same deep neural network. The variable $A_t \in \mathbb{R}^{T_s}$ is an attention vector that combines frame-wise features of the source utterance $X$ to generate the target frame $\hat{Y}_t$. Our model differs from standard sequence-to-sequence model by treating neural net predictions as residuals added to input sequence itself. Notice that the residuals depend on input and the history of predictions $\hat{Y}_{0:t-1}$ at previous time steps. This autoregressive property allows the neural network to learn segmental and supra-segmental variations that can potentially distinguish between the speakers or emotions.

During training, we use paired data $(X, Y)$ and maximize the likelihood of the target speech signal with respect to the neural network weights $\{\theta, \gamma\}$. This likelihood can be written as:

$$P(\hat{Y}, \hat{T}|X) = P(\hat{T}|X) \prod_{t=1}^{\hat{T}} P(\hat{Y}_t|X, \hat{T}, \hat{Y}_{0:t-1}), \tag{3}$$

where, the second term in Eq. (3) can be obtained via marginalization over $A_t$ as follows:

$$P(\hat{Y}_t|X, \hat{T}, \hat{Y}_{0:t-1}) = \sum_{A_t} P(\hat{Y}_t, A_t|X, \hat{T}, \hat{Y}_{0:t-1}, M)$$

$$= \sum_{A_t} P(\hat{Y}_t|X, \hat{T}, A_t, \hat{Y}_{0:t-1}) P(A_t|X, \hat{Y}_{0:t-1}, M) \tag{4}$$

The variable $M$ here denotes the attention mask. We introduce $M$ for convenience, as it is a deterministic function of the source length $T_s$ and the estimated target length $\hat{T}$. We encode the attention $A_t$ as a one-hot vector across the $T_s$ frames of the source speech. Thus, it follows a multinomial distribution. For simplicity, we model $A_t$ as conditionally independent of the target length $\hat{T}$ given the mask $M$ and the input $X$ (see Fig. 1). Taking $\log(\cdot)$ of Eq. (3) and combining with Eq. (4) yields:

$$\mathcal{L}(\theta, \gamma) = -\log\Big(\sum_{A_t} P(\hat{Y}_t, A_t|X, \hat{T}, \hat{Y}_{0:t-1}, M)\Big) - \log\big(P(\hat{T}|X)\big)$$

$$= -\log\Big(\sum_{A_t} \frac{q_\theta(A_t|X, \hat{Y}_{0:t-1}, M)}{q_\theta(A_t|X, \hat{Y}_{0:t-1}, M)} P(\hat{Y}_t, A_t|X, \hat{T}, \hat{Y}_{0:t-1}, M)\Big) - \log\big(P(\hat{T}|X)\big)$$

$$\leq -\sum_{A_t} q_\theta(A_t|X, \hat{Y}_{0:t-1}, M) \log\big(P(\hat{Y}_t|X, A_t, \hat{Y}_{0:t-1})\big) - \log\big(P(\hat{T}|X)\big) + KL(q_\theta(A_t)||P(A_t))$$

$$= -\sum_{A_t} q_\theta(A_t|X, \hat{Y}_{0:t-1}, M) \log\big(P(\hat{Y}_t|X, A_t, \hat{Y}_{0:t-1})\big) - \log\big(P(\hat{T}|X) - H(q_\theta) + const.$$

$$\leq -\sum_{A_t} q_\theta(A_t|X, \hat{Y}_{0:t-1}, M) \log \left( P(\hat{Y}_t|X, A_t, \hat{Y}_{0:t-1}) \right) - \log \left( P(\hat{T}|X) + const. \right) \quad (5)$$

The distribution $q_\theta(\cdot)$ above is an approximating distribution for the attention vectors implemented by a convolutional network. The first inequality uses the convexity of the $-\log$ function, and the second inequality comes from the fact that entropy $H(q_\theta) \geq 0$. Notice that we have implicitly assumed $P(A_t|X, \hat{Y}_{0:t-1}, M)$ has a uniform distribution over the masked region. This is a reasonable assumption given that the masking process reduces the attention domain to a small region (see Section 2.3). However, $q_\theta$ is **not penalized** for deviating from this uniform distribution during training. This flexibility allows the network to learn realistic attention vectors during autoregressive decoding. Eq. (5) can be easily translated into a neural network loss function which we minimize for $\{\theta, \gamma\}$:

$$\mathcal{L}(\theta, \gamma) = \lambda_1 \times E_{A_t \sim q_\theta} \left[ \log \left( P(\hat{Y}_t|X, A_t, \hat{Y}_{0:t-1}) \right) \right] + \lambda_2 \times \log \left( P(\hat{T}|X) \right)$$

$$= \lambda_1 \times E_{A_t} \left[ \|\hat{Y}_t - Y_t^0\|_1 \right] + \lambda_2 \times \|\hat{T} - T^0\|_1, \quad (6)$$

where $\lambda_1$ and $\lambda_2$ are the model hyperparameters that adjusts the trade-off between the two objectives and implicitly contain the variances of the Laplace distributions in Eq. (1). Notice that the loss in Eq. (6) computes an expectation over the attention maps. We use the Monte-Carlo estimate by sampling from the attention map at each time-step. The training procedure is therefore stochastic in nature due to this random sampling. We mix this stochastic version with the maximum aposteriori estimate (MAP) of the attention vector with a probability of 0.2 during the start of training procedure.

---

**Algorithm 1:** Strategy for model training

---

1   function trainModelParameters $(X, Y)$;
   **Input**   : filterbank energies $(X \in \mathbb{R}^{D \times T_s}, Y \in \mathbb{R}^{D \times T_t})$
   **Output:** model parameters $(\theta, \gamma)$
2   **if** $epoch < MaxEpochs$ **then**
3      Set $t = 0$, predict target length $\hat{T} = f_\gamma(X)$ and create the mask $M \in \mathbb{R}^{T_s \times T_t}$;
4      Estimate $A \in \mathbb{R}^{T_s \times T_t}$ using masked convolution and sample $u \sim U(0, 1)$;
5      **if** $u < 0.2$ **then**
6         Sample $a \in \mathbb{R}^{T_s \times T_t}$ as 1-hot vectors from $A$;
7         Reconstruct using $\hat{Y}_t = X \cdot a + f_\theta(X, Y_{0:t-1})$;
8      **else**
9         Reconstruct using $\hat{Y}_t = X \cdot A + f_\theta(X, Y_{0:t-1})$;
10      **end**
11      Compute prediction errors and update parameters $\theta, \gamma$;
12      epoch $\leftarrow$ epoch + 1;
13   **end**
14   return $\theta$ and $\gamma$;

---

## 2.2   Training and Testing Algorithm

Algorithm 1 describes our training strategy. First, we extract the filterbank energies from the paired input-output utterances and define the ground truth for length prediction as the ratio of the input-to-output sequence length. Next, we construct the attention mask (Sec. 2.3) for each input-output sample pair in the mini-batch. The neural network then processes the input frames and generates an embedding for the decoder operation and to predict the target sequence length. It further estimates an attention vector as a multinomial distribution at each decoder step inside the specified masked region. Finally, to compute the loss, we either sample from this multinomial distribution or directly apply the *maximum a posteriori* attention estimate to the input sequence to get the target frames.

We use a Bernoulli sampling procedure with probability of $0.2$ in line 5 (i.e., low contribution of the stochastic loss) to prevent the model from diverging in sub-optimal directions. The MAP estimate helps in this regard. Empirically, we found this to be extremely helpful in generating monotonic attention that is also group sparse in nature. We fixed the slope of attention mask in

---

**Algorithm 2:** Strategy for model testing (i.e., open-loop duration modification)

---

1 function modifyDuration $(X)$;
   **Input** : filter-bank energy ($X \in \mathbb{R}^{D \times T_s}$ and $Y_0$)
   **Output:** alignments $((x_1, y_1), (x_2, y_2), ...)$
2 Predict length of target sequence $\hat{T}_t = f_\gamma(X)$;
3 Create attention mask $M \in \mathbb{R}^{T_s \times \hat{T}_t}$ and Set $t = 0$;
4 **if** $t < \hat{T}_t$ **then**
5    Using mask $M_t$, $X$, and $Y_{0:t-1}$ estimate $A_t$;
6    Using $X$, $Y_{0:t-1}$, and $A_t$, predict $Y_t$;
7    $t \leftarrow t + 1$;
8 **end**
9 Run DTW backtracking on the attention matrix $A$;
10 return (alignments $(x_1, y_1), (x_2, y_2), ...(x_n, y_n)$);

---

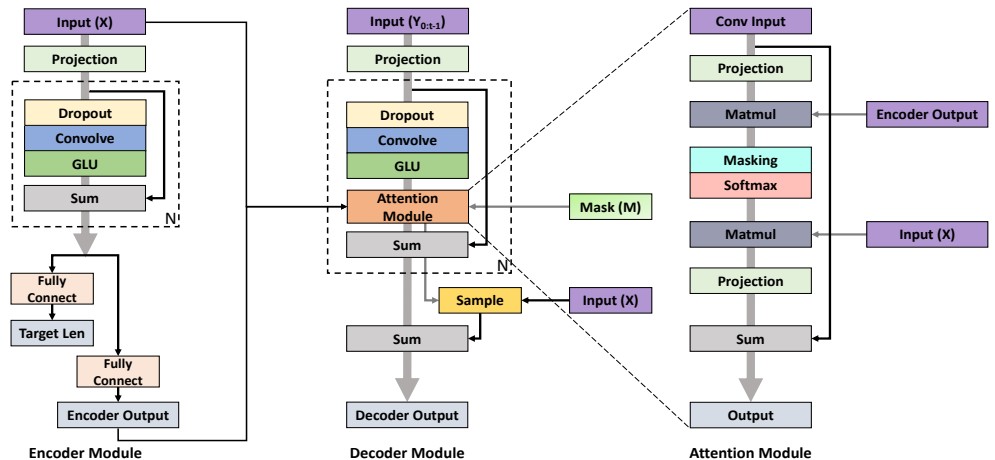

Figure 3: Neural network architecture used for the sequence-to-sequence speech generation. The encoder and decoder modules consist of 10 identical blocks. Projection layers are simple feed-forward layers without any non-linearity to project input features in high dimension.

line 4 to 1.25 based on the relative difference in length observed from the training datasets. The distribution $q_\theta$ is the variational distribution which is approximated by a fully convolutional neural network shown in Fig. 3. The testing procedure is shown in Alg. 2. Here, we do not sample from the attention distribution at each time step $t$. Instead, the encoder-decoder model is allowed to run autoregressively for the predicted number of time steps ($\hat{T}_t$). We use this attention map as a proxy for DTW similarity matrix to run Viterbi decoding strategy and obtain a sequence of co-ordinates representing the frame-wise correspondence between the input and estimated target speech.

The generative/testing procedure (Alg. 2) is similar to the training strategy except for some minor differences. In the generative mode, we use the predicted target sequence's length instead of the ground truth. Second, we do not sample from the attention vector as the estimated target sequence is no longer important. We use the attention map as a proxy for the DTW similarity/cost matrix for Viterbi alignment. This alignment allows us to rearrange the input frames to give modified speech.

### 2.3 MASKING

The mask $M$ is used to constrain the scope of the attention mechanism to be similar in time-scale to the input. This procedure is important for two reasons. From a speech quality perspective, large local swings in speaking rate may generate unintelligible speech. From an estimation perspective, the speech utterances contains hundreds (sometimes thousands) of frames. It is difficult to robustly train a deep network to generate such long attention vectors using smaller datasets.

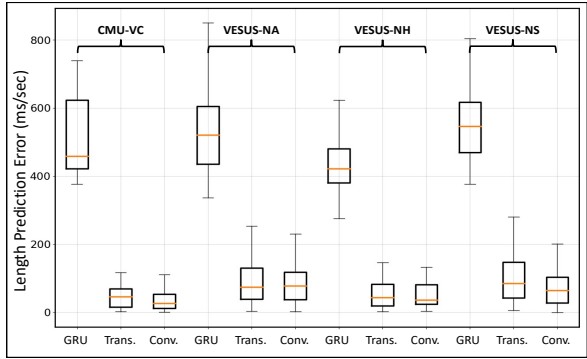

Figure 4: Comparing error in length prediction using encoder embeddings.

We use the masks derived from Itakura parallelogram (Itakura, 1975), as illustrated in Fig. 2. The Itakura parallelogram is generally used to speedup the DTW algorithm when the speaking rates in the source and target utterances are expected to lie in a certain interval. The slope of the Itakura parallelogram specifies the minimum and maximum speaking rates that the reconstructed utterances are allowed to possess in comparison to the input speech. In this paper, we fixed the minimum and maximum variation in speaking rate to $0.8$ and $1.25$, respectively after observing the training data.

## 2.4 NEURAL NETWORK ARCHITECTURE

We adapt the neural network architecture from (Gehring et al., 2017) by adding residual/skip connections to the last layer and reconfiguring the entire attention module. Fig. 3 shows the encoder, decoder and the modified attention module of the convolutional neural network used for experiments. The encoder is responsible for generating feature embeddings for the decoder and for predicting the relative length of target speech. The sample operation in Fig. 3 is responsible for generating a sample from the attention distribution required for reconstruction and backpropagation.

We train our model using mini-batch gradient descent and Adam optimizer (Kingma & Ba, 2015) with a fixed learning rate of $10^{-4}$. We fixed the batch size to 16. The input $X$ is 80-dimensional Mel-filterbank energies spanning 0-8kHz. The projection layer expands this input to 256 dimensions. Both the encoder and decoder consist of 10 convolutional layers, each followed by gated linear unit. Given the small size of training data, we use data augmentation to properly estimate the network parameters. Specifically, we reverse the input-output sequences and randomly extract intervals of variable size (with probability $0.5$) from the full speech utterance.

## 2.5 DTW BACK-TRACKING

Our final step uses the attention map produced by the encoder-decoder as a proxy for the DTW similarity matrix between the source and target speech frames. This strategy allows us to train the convolutional model on a relatively small dataset (e.g., 2-3 hours) and still generate intelligible speech during open-loop modification of new utterances. Effectively, we apply a dynamic programming operation to the attention maps produced by the neural network to get a path of alignment from source to target, rather than rely on the noisy spectral reconstruction for resynthesis (see Algorithm 2). To avoid skipping phonemes, we constrain the dynamic programming path to take at most one horizontal or vertical at a time while backtracking. Once estimated, the path informs a reorganization of the source utterance frames via local contraction dilation operations. Following this reorganization, the target speech is synthesized via the WORLD vocoder (Morise et al., 2016).

## 3 EXPERIMENTAL RESULTS

We evaluate our model on two multi-speaker datasets: CMU-ARCTIC (Kominek & W Black, 2004) and VESUS (Sager et al., 2019). We query various properties of the model on tasks described below.

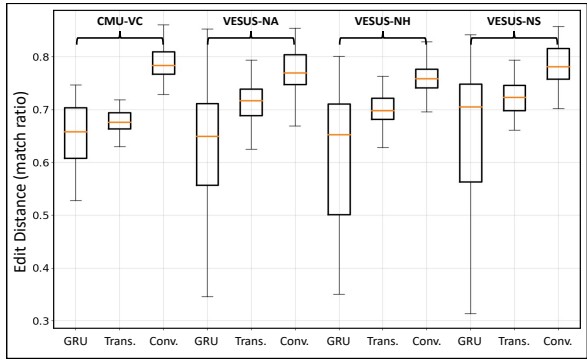

Figure 5: Comparing alignment similarity between attention map and DTW.

### 3.1 DATA AND VOICE MORPHING TASKS

CMU-ARCTIC database has 4 American English speakers (two male, two female), who we paired according to their gender, for voice conversion. This gives us a total of 2264 sentence pairs. We train our duration modification framework using 2164 utterances from the database and use the remaining 100 utterances (random 50-50 split) for validation and testing the open-loop modification properties.

VESUS is an emotional speech corpus containing 250 phrases read by 10 speakers in 4 emotion classes: neutral, angry, happy, and sad. Each utterance in VESUS corpus contains 10 crowd-sourced emotional annotation representing the saliency of intended emotion category. Given the variation in quality, we use only utterances that are correctly annotated by at least half of the listeners.

Our task on VESUS is to inject emotion into a neutral/monotone utterance. Thus, we train three duration models corresponding to the three neutral-emotional pairs, resulting in the following splits:

- **Neutral to Angry**: 2385 utterances for training, 72 for validation and, 61 for testing.
- **Neutral to Happy**: 2431 utterances for training, 43 for validation and, 43 for testing.
- **Neutral to Sad**: 2371 utterances for training, 75 for validation and, 63 for testing.

The utterances in VESUS corpus are short with an average signal duration ranging from $1.5 - 2$ seconds. Given the small sample size due to shorter length sequences, we train the convolutional neural network on CMU-ARCTIC from scratch and fine-tune it for each emotion conversion task.

### 3.2 LENGTH PREDICTION

As a sanity check, we compare the predicted utterance length by our framework with that of the ground truth parallel utterance. As seen in Fig. 3, the utterance length is predicted by summing the encoder embeddings across time and feeding the sum into a linear layer. At a high level, the summation aggregates information across the entire utterance without the instabilities of a recurrent architecture. Our baselines are two most commonly used sequence-to-sequence frameworks: (i) Gated Recurrent Unit or GRU model (Cho et al., 2014), and (ii) Transformer model (Vaswani et al., 2017). The architecture and training strategy of baseline models are described in more detail in the appendix section. Fig. 4 shows the error in predicting the length ratio in a ms/sec format. Notice that, our framework mispredicts the utterance lengths by only 40ms/sec and 65ms/sec (on average) on CMU-ARCTIC and VESUS, respectively. Duration prediction is particularly challenging on VESUS due to marked differences between neutral and emotional utterances. The median prediction error for GRU model is in the range of $400 - 600$ms per second of the input utterance. The Transformer model fares relatively well in comparison to GRU because of their ability to establish long-range dependency. However, our framework performs best in this challenging scenario, likely due to our multi-task setup and the fusion of deep representation with Bayesian regularization.

### 3.3 ATTENTION ALIGNMENT

Next, we compare the open-loop alignment between the source and target speech frames estimated via our attention map with the original DWT algorithm (where both utterances are known). To

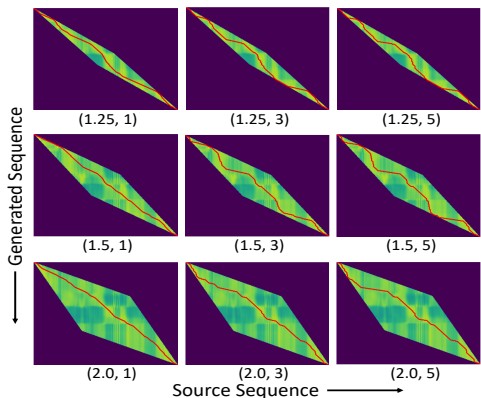

Figure 6: Effect of slope and step constraint on the alignment. The tuple under each image is in (slope, constraint) format. Red curves are the optimal path obtained via Viterbi back-tracking.

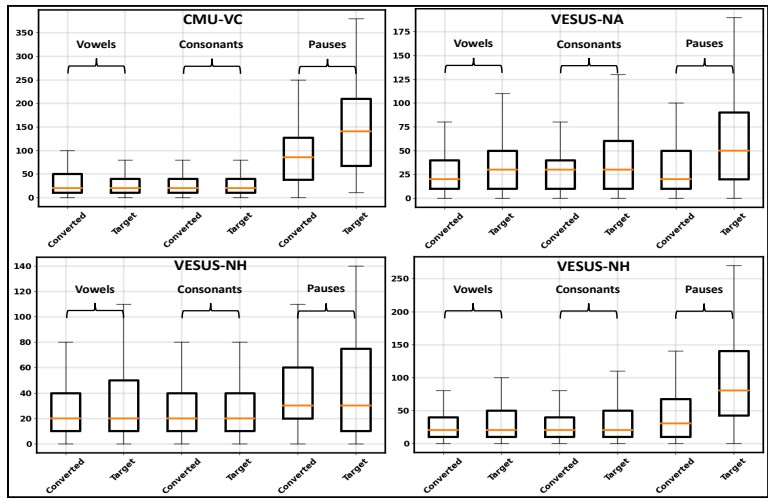

Figure 7: Comparing the duration difference between source/target and source/converted pair of utterances for vowels, consonants, and short pauses. Short pauses have large difference because our method does not generate silence frames by itself if it does not exist in the input utterance.

compare the warping paths, we code the horizontal, diagonal, and vertical moves of the backtracking procedure into three classes. We then compute the edit distance between the DTW alignment and the attention map based alignment. Once again, we compare our method against two popular sequence-to-sequence benchmarks. Fig. 5 illustrates the match ratio normalized by the average length of sequences. As seen, the match ratio varies between $0.70$ and $0.85$, which suggests that our approach is good at learning the general characteristics of duration modification. The GRU model performs poorly in this task due to its inability to learn sequence transformations in the order of $100$s of frames. The Transformer model does relatively well on this task as they can handle long sequences. The convolutional model however, performs best because of the Itakura masking constraint and its ability to exploit the continuity of short-time Fourier representation of speech. Furthermore, our method can be trained on limited data resources. In contrast, sequence-to-sequence models for voice conversion require hundreds of hours of training data along with sophisticated noise removal models to generate actual speech. Thus, our method can be used as a tool for manipulation of speaking rate at both, local and global scale. To our knowledge, this is the first demonstration of its kind.

Fig. 6 shows the effect of modifying the slope of Itakura parallelogram and the horizontal/vertical movement constraint imposed during the dynamic decoding stage. As expected, relaxing the slope constraint and increasing the number of consecutive horizontal/vertical moves provide more flexibility in adjusting to the speaking rate of generated speech. However, this flexibility can lead to

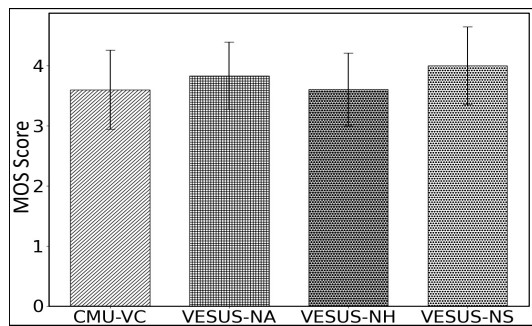

Figure 8: MOS of speech generated by our model evaluated via AMT.

missing or distorted phonemes, suggesting a trade-off between changing the speaking rhythm and preserving naturalness. We allow the users to tune these knobs for their own end application.

### 3.4 COMPONENT-WISE DURATION ANALYSIS

To further probe into the behavior of our method, Fig. 7 compares the difference in duration between our converted utterances and the ground truth target utterances for vowels, consonants and short pauses (e.g., silence and unvoiced phonemes). We use Penn phonetic forced alignment tool (Yuan & Liberman, 2008) to get the location and duration of each phoneme in a given utterance and group them into the above three categories. The results indicate that our proposed method faithfully modifies the duration of vowels and consonants. However, it is less effective with short pauses, as seen across all four tasks. This trend is intuitive, as we cannot create pauses if these frames do not exist in the source utterances. One direction of future work is to add a branch to our deep network that estimates the duration of short pauses intermittently within the utterance. Nonetheless, our model consistently estimates the difference between vowel and consonant duration across multiple tasks, which corroborates our claim of developing a general purpose speech rate manipulation framework.

### 3.5 SPEECH RECONSTRUCTION QUALITY

Finally, we use crowd sourcing to obtain a mean opinion score (MOS) for the re-synthesized speech quality of the testing utterances. The crowd sourcing was performed using Amazon mechanical turk (AMT). We collect 5 listener ratings for each converted utterance in the test set to leverage the sample averaging effect for a reliable estimate of MOS. Further, we also add some noisy utterances to the converted samples set to flag non-invested listeners and bots on AMT. As seen in Fig. 8, our method achieves an average MOS between $3.7 - 4.0$ across the four tasks. This performance is on par with the speech quality produced by state-of-the-art neural vocoders such as Oord et al. (2016); Shen et al. (2017); Kalchbrenner et al. (2018). We note that CMU-ARCTIC task has the lowest MOS, perhaps due to the longer and more complex utterances. Interestingly, the MOS is unaffected by errors in length prediction, as evidenced by the VESUS neutral-angry emotion conversion task. This suggests that our approach of combining the neural network attention weights with a structured DTW algorithm provides robustness to both the speech characteristics and estimation errors.

### 4 CONCLUSIONS

We have presented a novel deep-generative framework for adaptive speech duration modification. Our model used a convolutional encoder-decoder architecture to estimate attention maps to associate frames of the input speech with frames of the target speech. The attention maps are modeled as latent variables, which lead to a stochastic formulation of the mean absolute error (MAE) loss for model training. During testing, the attention map is directly used to approximate the similarity matrix for a DTW-style backtracking procedure. We evaluated our framework on a voice conversion and three separate emotion conversion tasks. Overall, our framework produced similar duration modification as the vanilla DTW, but *without requiring access to the target utterance*. Further, we showed that the re-synthesized speech had similar naturalness to most state-of-the-art neural vocoders.

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

# A APPENDIX

## A.1 DESCRIPTION OF GATED RECURRENT UNIT (GRU) MODEL

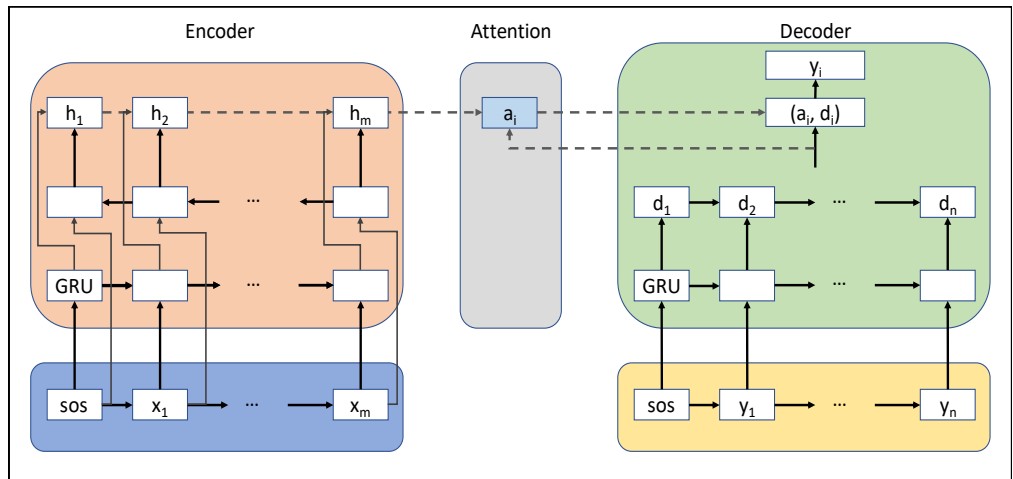

Figure 9: GRU model architecture.

We compare the performance of our method with existing state-of-the-art techniques for sequence-to-sequence conversion. RNN-LSTM frameworks have been a benchmark for many natural language processing tasks. They have only been sparingly used for speech conversion tasks due to longer sequence length coupled with the inability of RNNs to remember distant past. For our baseline, we choose GRU (gated recurrent unit) cells (Cho et al., 2014) based encoder-decoder model due of their low computational footprint. Fig. 9 shows the architecture of the GRU model for length prediction and approximating the DTW similarity matrix. The sum of encoder embeddings across time-axis is fed into a linear layer for target length prediction. The encoder and decoder in the GRU model consist of a single layer of $64$ cells running in both, forward and backward directions. They are further connected via an attention layer implemented by a single feed-forward layer. We train the model using Adam optimizer with default parameters in PyTorch (Paszke et al., 2017) and a fixed learning rate of $1e-5$. We follow the same data augmentation strategy as the proposed approach but remove the masking constraint for the attention alignment to stay close to the vanilla model.

## A.2 DESCRIPTION OF TRANSFORMER MODEL

We also train a vanilla Transformer model from scratch for the voice conversion task and then fine-tune it for emotion conversion. The Transformer model has 6 layers of encoder and decoder and each layer has 4 attention heads. We use the same data augmentation procedure as our proposed technique and train it via Adam optimizer with default PyTorch settings. Similar to the GRU model, there is no masking for the encoder-decoder attention module. We use the sum of encoder embeddings across time-axis to predict the target length sequence. Transformer models can handle longer inputs due to their non-sequential nature and ability to form long-range dependencies between input tokens. Therefore, they naturally perform well on estimating the length of target sequence by learning appropriate encoder representation. The Transformer model however, fails to leverage the continuity in the short-time representation of speech signal. An additional constraint we implicitly imposed on the baselines and proposed neural network architecture is their ability to run on a single NVIDIA K80 GPU. We are able to fit multiple layers of convolutional and Transformer model but only single layer of GRU on this GPU. It allowed us to be more innovative with our approach.

## A.3 ABLATION ANALYSIS: REMOVING ITAKURA MASKING

There are multiple components in the proposed model which work in synchronization to produce naturally sounding speech. In addition to the generative modeling part, the two most important augmentations we have made to the standard deep neural network pipeline are: (i) using Itakura

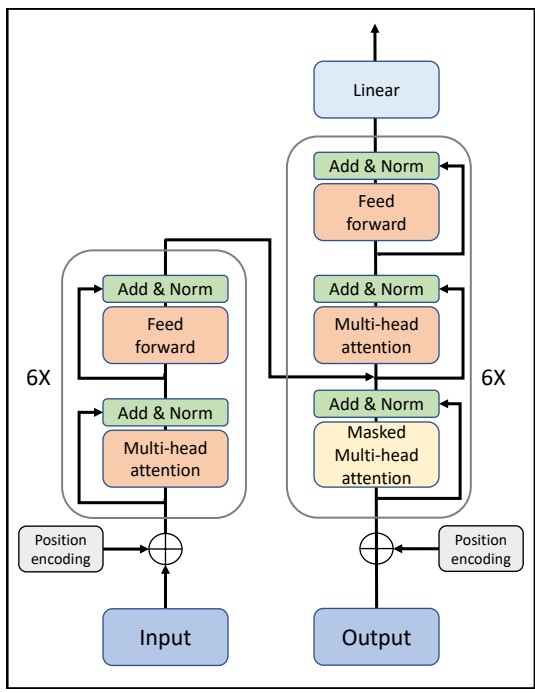

Figure 10: Transformer model architecture.

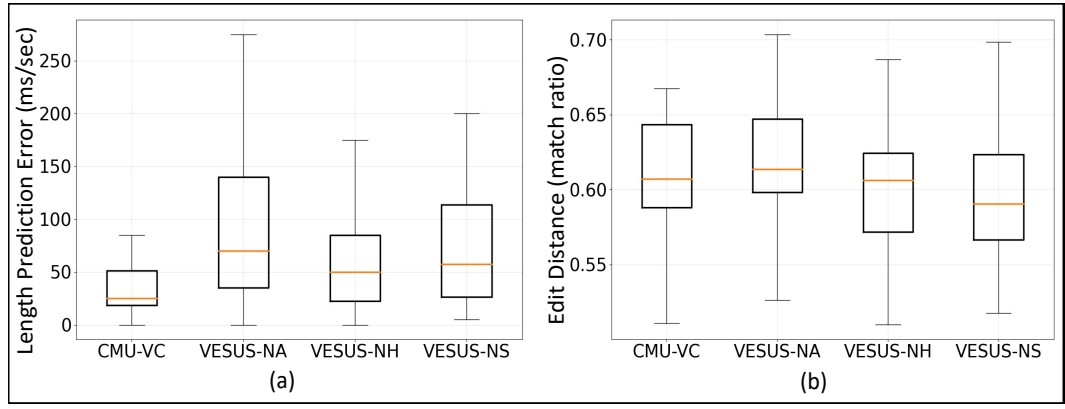

Figure 11: (a) Length prediction of target utterances and (b) measuring similarity of attention map to DTW cost matrix. The model is trained without masking constraint imposed in the attention layer.

masking for attention scope and (ii) using an attention weighted residual connection in the final layer. Therefore, we perform ablation experiments to understand the relative significance of each of these augmentations. Our first experiment removes masking from the attention layers. Fig. 11 shows the model's performance on target length prediction and approximating the DTW similarity matrix. The results in Fig. 11(a) indicate that the length prediction performance is roughly similar to the proposed model. This is expected because, we hypothesized that it is the multi-task setup that allows the convolutional network to estimate length with a relatively small error (in ms/sec). The match ratio metric however, (shown in Fig. 11(b)) is considerably worse. Itakura masking procedure acts as a good inductive bias/prior on the attention map because the speech rate do not fluctuate drastically in human conversations. Therefore, the attention map is localized in our model.

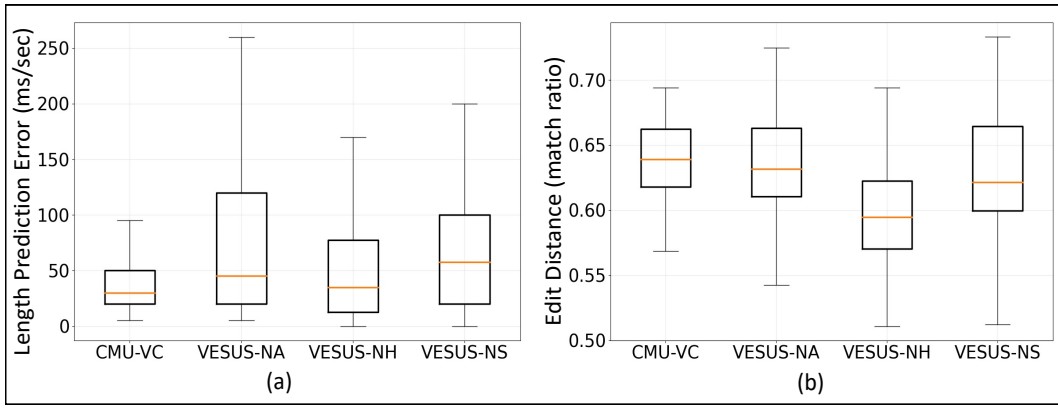

Figure 12: (a) Length prediction of target utterances and (b) measuring similarity of attention map to DTW cost matrix. The model is trained without residual connection in the final decoder layer.

## A.4 ABLATION ANALYSIS: REMOVING RESIDUAL CONNECTION

Our second ablation experiment involves removing the attention weighted residual connection from the final layer of decoder. Fig. 12(a) shows that the model is able to estimate the target sequence lengths with a relatively low error rate. This is again due to multi-task training setup. The match ratio (Fig. 12(b)) in this experiment is better than the no-masking results but, worse than the proposed model. Therefore, we can confidently say that Itakura masking helps in approximating DTW similarity matrix. Further, the presence of residual connection is extremely important as it provides a good starting point for the convolutional network to start predicting the target frames. Since the linguistic content of input and target utterances are same, the residual connection allows the neural network to inherit input speech properties which is helpful in auto-regressive generation mode.

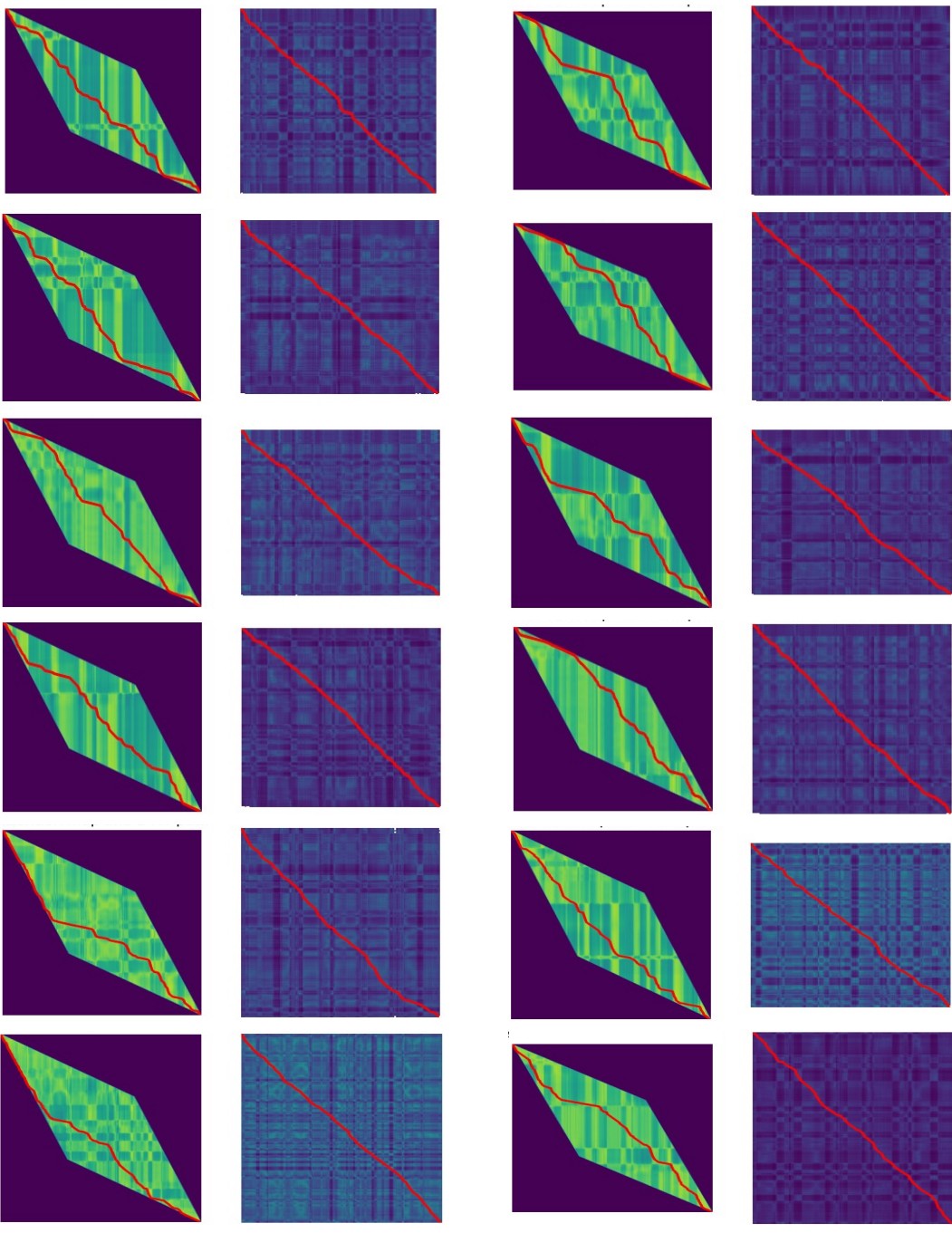

Figure 13: Examples of alignment path obtained using encoder-decoder attention map (left) and ground truth DTW backtracking procedure (right). The source sequence lies on the x-axis and the target/generated sequence lies on y-axis. Note that, unlike DTW, the convolutional neural network does not use the ground truth target utterance. The attention map and the DTW similarity matrix exhibits similar block structure arising due to the short-term stationarity of speech signals.

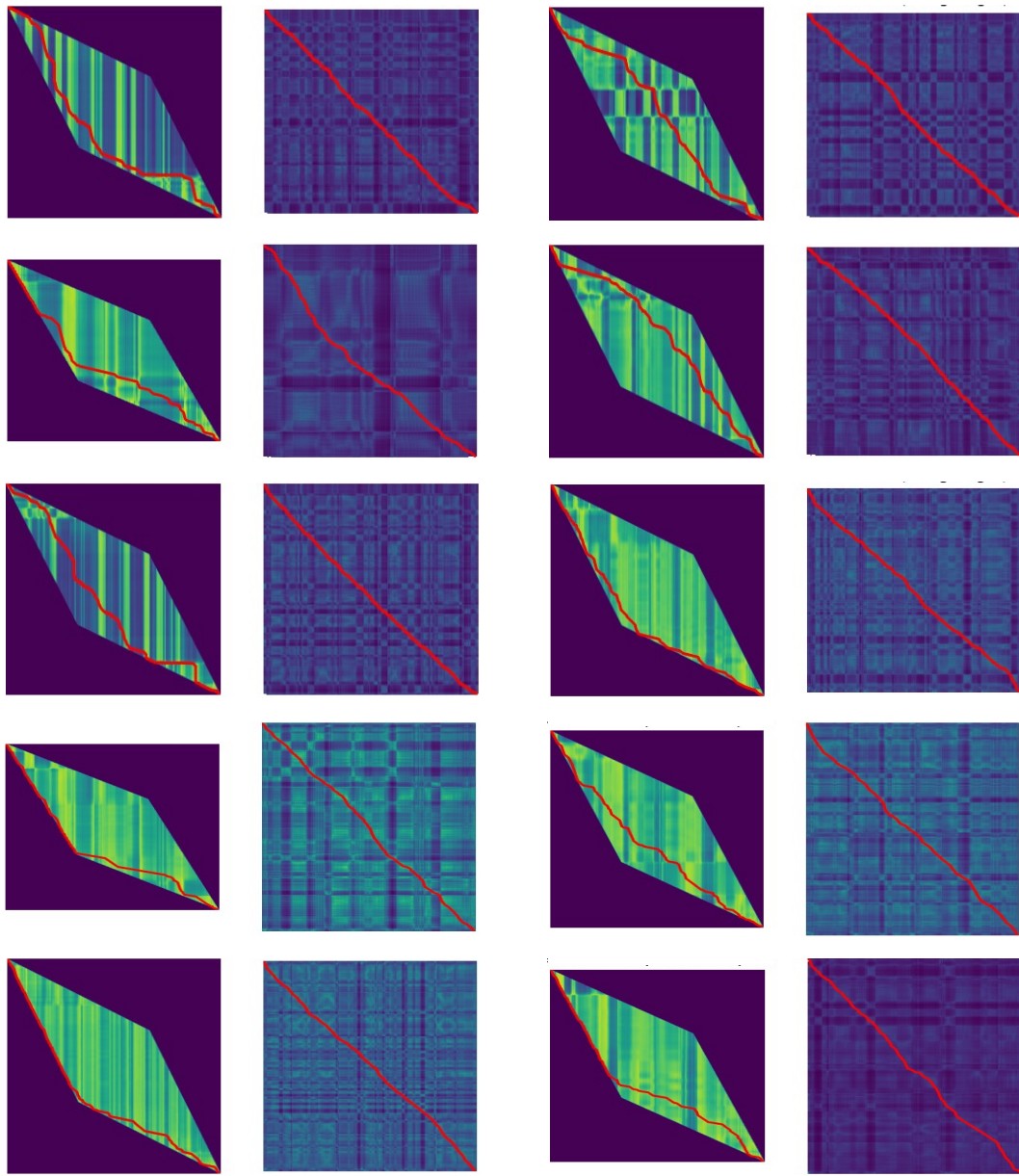

Figure 14: Examples of alignment path obtained using encoder-decoder attention map (left) learned **without masking constraint** and ground truth DTW backtracking procedure (right). The source sequence lies on the x-axis and the target/generated sequence lies on y-axis. The attention map no longer exhibits the block structure and is distributed uniformly across the frames of source sequence.

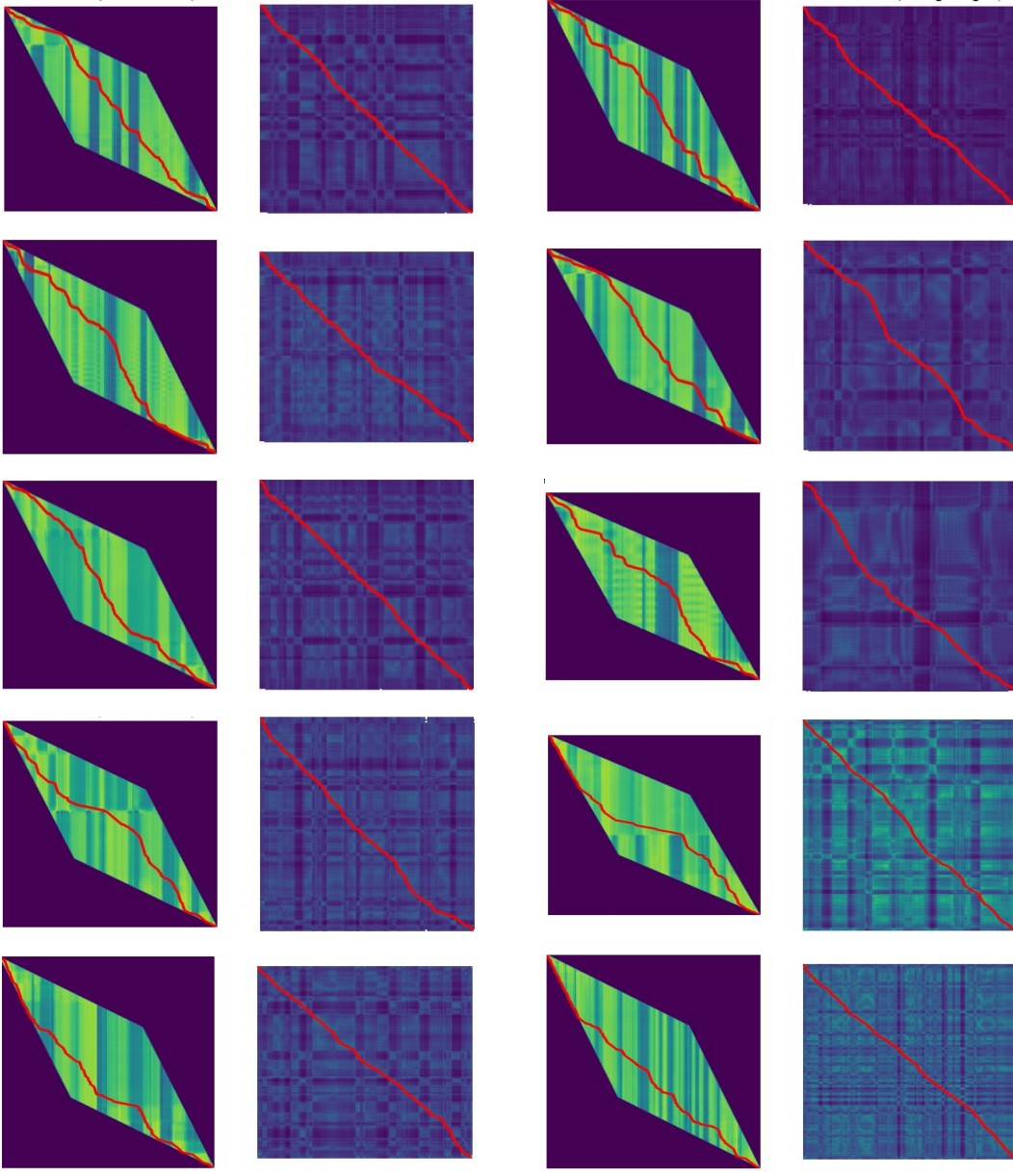

Figure 15: Examples of alignment path obtained using encoder-decoder attention map (left) learned **without residual connection** and ground truth DTW backtracking procedure (right). The source sequence lies on the x-axis and the target/generated sequence lies on y-axis. Once again, the attention map no longer exhibits the block structure and is distributed uniformly on y-axis in most cases.

