# OpenReview forum: "Adaptive Speech Duration Modification using a Deep-Generative Framework"
_ICLR.cc/2022/Conference — ICLR 2022 Submitted_

### Official Review · Reviewer_Lbse · 2021-10-31

**Correctness:** 2
**Technical Novelty And Significance:** 2
**Empirical Novelty And Significance:** 1
**Recommendation:** 1
**Confidence:** 4

**Details Of Ethics Concerns:**

In the notion of voice/emotion conversion, it may pose risk to individuals/celebrities since the sentiment of the speech could be manipulated, even if the text being read is not modified. However, the authors haven't written the "Ethical considerations" section to address those issues.

**Main Review:**

The proposed method looks interesting and novel, but there are several concerns over the evaluation, correctness, and clarity.

### Evaluation

- The performance of the proposed method is presented by reporting (1) length prediction error, (2) alignment similarity between the attention map and the original DTW result, and (3) MOS of re-synthesized speech.
- A major concern over the evaluation is the absence of a proper baseline method, which is crucial for presenting the motivation of the work and the contribution of the paper. This problem applies to all 3 metrics and must be addressed by comparing them with existing works or the simplified version of the proposed method. Replacing the convolution architecture with GRU or Transformers does not look like a proper ablation, since the architecture itself is irrelevant to the proposed generative model.
- Regarding the length prediction error, the reviewer considers this as an inappropriate metric for generative models on speech. Speech is inherently stochastic - meaning that the duration of the speech can vary even when the text and emotion are kept identical. Recent works on speech synthesis [1, 2] aim to make the duration of generative speech as stochastic as possible. Thus, the reviewer recommends the authors critically think about whether the length prediction error is a suitable metric for the proposed method.
- Please elaborate on an alignment similarity metric, since the readers won't be able to judge how high/low the values given in Figure 5 are.
- Providing the audio samples for the generative models on speech is considered necessary to convince the readers. Please supplement them.

### Correctness

- In section 3.3, the authors state that: "In contrast, the sequence-to-sequence models for voice conversion requires hundreds of hours of training data along with sophisticated noise removal models to generate actual speech." The reviewer strongly disagrees with this statement. Recent state-of-the-art works on voice conversion [3] can work with an amount of data that does not differ much from VESUS. It is also unfair to directly compare with previous works since the proposed method has utilized data augmentation, as stated in section 2.4.
- In section 3.5, the authors state that the speech quality from the proposed method matches that of WaveNet, Tacotron2, and WaveRNN since the average MOS was 3.7 – 4.0. But, this is not correct. Since the MOS results of a single model widely vary across the experiments and the dataset, the MOS of the existing models should be re-measured using the same evaluation protocol to be directly compared with the proposed method.
- Since the proposed method is working on manipulating the mel-spectrograms, it is basically an acoustic model. Hence, in order to fairly compare the speech quality with existing models, it should not be directly compared with vocoders. For instance, the author might want to use a single vocoder for all models (including baseline).

### Clarity

- There are some unclear parts that might make the reader hard to comprehend the paper. To be specific:
  - In section 2.2, what does the following sentence mean? "The neural network then processes the input frames and generates an embedding for the decoder operation and to predict the target sequence length." It would be clearer if the referral to the specific line(s) from Algorithm 1 was made.
  - In Algorithm 1, what is a variable 't' in lines 7, 9?
  - The location of Figure 2 (start of page 3) is too far away from the section that the reference is made.
- Some minor mistakes that did not affect the reviewer's judgement include (please fix):
  - Strange referencing: "(dtw, 2008)"
  - Unclosed parentheses in 4th, 5th line of equation 5
  - Typo in section 3.3: DWT -> DTW

Some general advice on paper's clarity: Figures do not speak for themselves. The authors might want to add some elaborations on each figure's caption to state what the takeaways are, even if it overlaps with the main text.

### Overall comment

The reviewer would like to appreciate the amount of work done in this work, however, several issues exist and must be addressed to be presented at ICLR. Thus, the reviewer (initially) recommends rejecting the paper.

### References

- [1] Rafael Valle, Kevin Shih, Ryan Prenger, and Bryan Catanzaro, "Flowtron: an autoregressive flow-based generative network for text-to-speech synthesis.", ICLR 2021.
- [2] Kim, Jaehyeon, Jungil Kong, and Juhee Son. "Conditional Variational Autoencoder with Adversarial Learning for End-to-End Text-to-Speech." arXiv preprint arXiv:2106.06103 (2021).
- [3] Z. Yi, W.-C. Huang, X. Tian, J. Yamagishi, R.K. Das, T. Kinnunen, Z. Ling, T. Toda, "Voice Conversion Challenge 2020 -- intra-lingual semi-parallel and cross-lingual voice conversion --" Proc. Joint workshop for the Blizzard Challenge and Voice Conversion Challenge 2020, pp. 80-98, Oct. 2020.


**Summary Of The Paper:**

This work proposes a generative model to convert emotion by adaptively modifying the duration of speech components without the need for reference utterance. The proposed method is evaluated via measuring (1) length prediction error, (2) alignment similarity between the attention map and the original DTW result, and (3) MOS of re-synthesized speech.

**Summary Of The Review:**

The reviewer vote for rejecting the paper; several issues on evaluation, correctness, and clarity are visible.

---

### Official Review · Reviewer_rjMF · 2021-11-02

**Correctness:** 2
**Technical Novelty And Significance:** 3
**Empirical Novelty And Significance:** 2
**Recommendation:** 3
**Confidence:** 4

**Details Of Ethics Concerns:**

1. Reference to the source code location compromises the author identity.
2. Pilot study of users using a crowd-sourcing platform.

**Main Review:**

Strengths
1. A thorough understanding of the problem place (speech parameter modification).
2. Reexamination of an important and practical problem using recently advances in transformers.
3. Evaluation on 2 public domain datasets.
4. Pilot user study.


Things to consider
1. This statement, as it is stated in the abstract is misleading/incorrect. "We propose the first method to adaptively modify the duration of a given speech signal." This relies on a definition of adaptation that is only available to the reader later.
2. There are lots of strong claims in the Introduction that could be avoided. Drawbacks of DTW and neural vocoders are unwarranted without explicitly stating the practical applications goals being considered by the authors. The statement on data needs is misleading. For example, the present work derive key parameters, such as the slope of the attention mask, based on the training data.
3. This statement is confusing. "We mix this stochastic version with the maximum aposteriori estimate (MAP) of the attention vector with a probability of 0.2 during the start of training procedure." Do you mean a weighted additive approach?
4. Lots of hand-wavy details such as "Empirically, we found this to be extremely helpful in generating monotonic attention that is also group sparse in nature."
5. Small-subject MOS-based comparisons across publications are problematic due to uncalibrated listener expertise. I appreciate the authors' efforts in conducting an user study but conclusions must be more guarded.
6. The paper references only very early work on pitch and rate modification and a few recent ones. A couple of decades of work after PSOLA (90s - 2005) could be added to provide a more comprehensive review for the reader.

**Summary Of The Paper:**

Speech rate, duration, and pitch modification is of interest in several practical audio applications. The paper proposes the use of an encoder-decoder framework with attention masking to estimate a candidate target utterance length to overcome the need for a priori knowledge of a target utterance on a speaking rate modification task. Evaluation on 2 standard databases show that the proposed approach performs better than 2 sequence-to-sequence models that were originally developed for different domains.

**Summary Of The Review:**

The paper brings the state-of-the-art ML research to an important but less active filed of study: speech signal modification. The proposed approach is reasonable and evaluations are solid. However, the authors haven't done a great job of explaining the drawbacks of previous approaches, in practice, and the critical need for unsupervised adaptation. Hence, the overall paper reads as a solution (transformer-based models) looking for a new application. The comparison baselines also reflect this discrepancy. A better comparison would be also related approaches in pitch and style modification.

---

### Official Review · Reviewer_EmGa · 2021-11-05

**Correctness:** 3
**Technical Novelty And Significance:** 3
**Empirical Novelty And Significance:** 3
**Recommendation:** 6
**Confidence:** 4

**Main Review:**

The introduction glosses over most of the prior work in this area. The authors write: "comparatively little progress has been made in changing the duration/speaking rate of an utterance" -- Perhaps the authors have neural/deep models in mind, or some other narrow claim in mind, but time-scale modification is a classical topic in speech and signal processing. The intro, however, jumps from the TD-PSOLA approaches of the mid 80s to a 2017 reference, and skips over intervening work, like sinusoidal-based approaches (to cite one). (I'm not including DTW in this view, since it's not a true time-scale modification algorithm and requires knowledge of target durations.) A better literature review is needed.

Re ealier approaches: "the user must manually specify both the portion of speech to modify and the exact manner in which it should be altered. Hence, the method is neither automated nor adaptive" -- One could argue that these approaches are, in fact, more flexible since they allow full (and local) control over what parts of speech to modify, a feature that made them a building block for many years of speech synthesizers that needed precise modification of specific speech segments. They suffer from many drawbacks (e.g., perceptual audible artifacts) that deep approaches can alleviate, and models that allow implicit duration modification (as in this work) are very attractive, so I think it shouldn't be hard to build better arguments to justify the value of this contribution.

"In this paper, we introduce the first fully-automated adaptive speech duration modification scheme" -- Existing time-scale algorithms are adaptive and automatic. Perhaps this claim needs to be written in narrower terms to be true (that doesn't require external duration specification?, etc.), but even then it's hard to evaluate it in the absence of a solid prior work comparison.


"Furthermore, our method can be trained on limited data resources. In contrast, sequence-to-sequence models for voice conversion require hundreds of hours of training data along with sophisticated noise removal models to generate actual speech." -- I perceive 2 issues with this claim: One is that it seems exaggerated. There will be a natural trade-off with quality, but it doesn't really require hundreds of hours of speech to train a voice-conversion system between, say, a small number of voices (which is the case the authors tackle in the evaluation). And secondly, these bigger voice-conversion systems are tasked with performing a far more complex objective than what the authors have proposed to do, and evaluated, in this submission, namely duration modification. We don't know from this evaluation whether the proposed model has captured any of the target-speaker characteristics, so the comparison between models that are solving different tasks seems unfair.

The subjective evaluation is somewhat limited by the small number of independent ratings (5) per sample. "This performance is on par with the speech quality produced by state-of-the-art neural vocoders" -- We shouldn't compare results across independent listening tests, but generally we should expect the SotA neural vocoders mentioned to outperform a WORLD vocoder. I wonder what the  results would look like if the authors had chosen to include the natural, unmodified, speech in the mix of samples to provide a calibrating topline performance.

Clarifications Needed:

-- Is there a point-wise or matrix-vector multiplication in Eqn. 2 between X and A_t? Looks like the former, but X is in R^{DxT_s} and A_t is in R^{T_s}

-- I'm confused by the acoustic features the authors predict with the network ("In this work, X corresponds to the Mel filterbank energies") and the fact that the samples are synthesized using the WORLD vocoder (which requires other features like pitch, spectral envelope, aperiodicity).


Minor edits:
** Notice that, our framework --> Notice that our framework

** Y \in R^{DxT} (in 2.1) --> Elsewhere in the paper this is R^{DxT_t}



**Summary Of The Paper:**

This paper proposes a generative sequence-to-sequence encoder-decoder architecture with attention for modifying the length of an input sequence. The authors derive the training loss for learning the network parameters from a principled Bayesian formulation and a variational inference bound, and use this to train the model from paired {input, output} sequences. During inference, a task-specific network is capable to estimating a target length, and modify the signal accordingly. It would seem that the architecture is capable of handling other transformations, and in fact the authors apply the model to speaker- and emotion-morphing tasks, but the bulk of the evaluation is on how accurate the  target length is estimated, and (briefly) the perceptual quality of the resulting samples.


**Summary Of The Review:**

The paper provides a novel contribution by bringing neural generative models to the task of time-scale modification of speech signals. The work is evaluated in terms of objective metrics and shown to outperform some baselines. A (somewhat limited) listening test suggests this model attains high-quality samples though there may be some limitations in interpreting these results. The work is well written, but some of the claims need to be revised, and the contributions better situated against prior work.

---

### Official Review · Reviewer_uzK2 · 2021-11-06

**Correctness:** 4
**Technical Novelty And Significance:** 2
**Empirical Novelty And Significance:** 2
**Recommendation:** 3
**Confidence:** 2

**Main Review:**

Strength:
- Very clear, easy to read paper
- Interesting combination of assumptions for loss upper bound derivation

Weakness:
- Some of the modeling choices are not very clear:
1. Why Laplace distribution was chosen for length T and target Y_t in Eq1
2. Conditional independence of output length and A_t given mask M and input X: I can see it is needed for derivation
of the upper bound (Eq 4), but I am not completely convinced if it is a realistic assumption


**Summary Of The Paper:**

This paper proposes a model for adaptive duration modification of an input signal. The model is a graphical model with neural components. By making some assumptions, authors derived an upper bound for the likelihood (conditional probability of output sequence and estimated length given input). All the model parameters are updated, by minimizing this upper bound.

**Summary Of The Review:**

My main concern with this paper is its scope, novelty and impact.
From modeling and formulation perspective, I do not see much novelty. There is not a novel modeling paradigm which be of interest of general machine learning crowd (as a reader I did not learn a new ml method). Of course, the paper proposes a novel method by combining some existing machine learning tools (like graphical models, sequence-to-sequence modeling). In my opinion, this novelty could have been justified if the method was applied to some high impact problem. Unfortunately, it is not clear to me if the speech duration modification is such a problem.

---

### Decision · Program_Chairs · 2022-01-20

**Decision:**

Reject

**Comment:**

This paper introduces a deep neural network sequence-to-sequence framework for modifying the length of a speech sequence.  It employs a convolutional encoder-decoder architecture optimized under a Bayesian formulation with variational inference.  The proposed framework is evaluated on a voice conversion task and three emotion conversion tasks. The results show that it can successfully change the duration of an utterance without accessing the target utterance.  Almost all reviewers raised concerns with some strong or inaccurate claims made by the authors in the paper.  The literature review on related work also needs to be significantly improved.  Another major concern is on experiments. Other than the DTW compared in the work, the proposed method should also be compared with existing duration modification techniques. The MOS evaluation seems to be limited and needs further improvement to make the results stronger and more convincing.  Since the authors did not provide a rebuttal, all these major concerns remain unanswered.